# MelanoGANs: High Resolution Skin Lesion Synthesis with GANs

**Christoph Baur**
Computer Aided Medical Procedures (CAMP)
TU Munich, Germany
c.baur@tum.de

**Shadi Albarqouni**
Computer Aided Medical Procedures (CAMP)
TU Munich, Germany

**Nassir Navab**
Computer Aided Medical Procedures (CAMP)
TU Munich, Germany
Whiting School of Engineering
Johns Hopkins University, Baltimore, United States

## Abstract

Generative Adversarial Networks (GANs) have been successfully used to synthesize realistically looking images of faces, scenery and even medical images. Unfortunately, they usually require large training datasets, which are often scarce in the medical field, and to the best of our knowledge GANs have been only applied for medical image synthesis at fairly low resolution. However, many state-of-the-art machine learning models operate on high resolution data as such data carries indispensable, valuable information. In this work, we try to generate realistically looking high resolution images of skin lesions with GANs, using only a small training dataset of 2000 samples. The nature of the data allows us to do a direct comparison between the image statistics of the generated samples and the real dataset. We both quantitatively and qualitatively compare state-of-the-art GAN architectures such as DCGAN and LAPGAN against a modification of the latter for the task of image generation at a resolution of 256x256px. Our investigation shows that we can approximate the real data distribution with all of the models, but we notice major differences when visually rating sample realism, diversity and artifacts. In a set of use-case experiments on skin lesion classification, we further show that we can successfully tackle the problem of heavy class imbalance with the help of synthesized high resolution melanoma samples.

## 1 Introduction

Generative Adversarial Networks (GANs) (6) have heavily disrupted the field of machine learning. In the computer vision community, they have been successfully used for the generation of realistically looking images of indoor and outdoor scenery(17; 4), faces (17) or handwritten digits (6). Their conditional extension (14) has also set the new state-of-the-art in the realms of super-resolution (11) and image-to-image translation (8). Some of these successes have been translated to the medical domain, with applications for cross-modality image synthesis (19), CT image denoising (20) and even for the synthesis of biological images (16), PET images (2), prostate lesions (10) and OCT patches (18).

The synthesis of realistic images opens up various opportunities for machine learning, in particular for the data hungry deep learning paradigm: Since deep learning requires vast amounts of labeled training data, which is often scarce in the medical field, realistically looking synthetic data may be used to

1st Conference on Medical Imaging with Deep Learning (MIDL 2018), Amsterdam, The Netherlands.

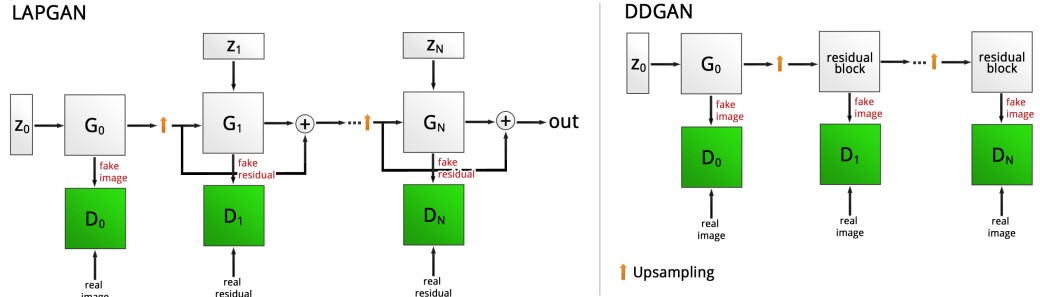

Figure 1: A comparison between the LAPGAN(left) and our DDGAN(right) architecture. While the LAPGAN focuses on the generation of realistically looking residual images from multiple sources of noise, our DDGAN directly optimizes for real images instead, while implicitly learning the residuals from a single source of noise.

increase the training dataset size, to cope with severe class imbalance and to potentially improve robustness and generalization capability of the models. Successful attempts for data augmentation using GANs have been made in (1; 5). Additionally, a trained GAN can provide valuable insights into the latent structure behind data distributions, e.g. by investigating the connection between the latent manifold and the generated images (21), ultimately facilitating data simulation.

GANs have been successfully used for various image synthesis tasks. Thoroughly engineered architectures such as DCGAN (17) or LAPGAN (4) have proven to work well for high quality image synthesis, however at resolutions of 64x64px and 96x96px, respectively, as reported in the original papers. In the context of fine-grained image classification, recent work (15) has pointed out the importance of high resolution data: The authors reported that using synthetic 32x32px images upsampled to 128x128px rather than using synthetic 128x128px images right away leads to a noticable decrease in classifier performance, which clearly motivates the generation of realistically looking images at higher resolutions.

**Contribution** In this work, we aim to increase the resolution of generated images while maintaining high quality and realism. For our experiments, we choose the ISIC 2017 dataset (3), consisting of approx. 2000 dermoscopic images of benign and malignant skin lesions. The nature of the images in the dataset allows us to directly compare the image statistics of both the real and the generated data. For data generation, we employ state-of-the-art architectures such as DCGAN or LAPGAN and rank them against a modification of the latter. More precisely, in contrast to LAPGAN, which involves multiple sources of noise, we experiment with a single source of noise, a discrimination on real and synthesized images rather than residuals and further try to learn an upsampling instead of using traditional interpolation. This leaves us with a network which can be trained end-to-end and has multiple discriminators attached to different levels of the generator, thus we refer to it as the deeply discriminated GAN (DDGAN). A comparison in terms of the aforementioned image statistics shows that all of the models match the training dataset distribution very well, however visual exploration reveals noticable differences in terms of sample diversity, sharpness and artifacts. In a variety of use-case experiments for skin lesion classification we further show that synthetic high resolution skin lesion images can be successfully leveraged to tackle the problem of severe class imbalance.

The remainder of this manuscript is organized as follows: We first briefly recapitulate the GAN framework as well as the DCGAN and the LAPGAN before we introduce our proposed DDGAN architecture. This is followed by an experiments section, where we try to synthesize realistically looking skin lesion images at a resolution of 256x256px using DCGAN, LAPGAN and different instances of the DDGAN. In the second part of the experiments section, we compare the performance of a state-of-the-art skin lesion classifier trained in the presence of severe class imbalance against models where the class imbalance has been resolved with the help of synthetic images.

## 2  Background

### 2.1  Generative Adversarial Networks

The original GAN framework consists of a pair of adversarial networks: A generator network G tries to transform random noise $z \sim p_z$ from a prior distribution $p_z$ (usually a standard normal distribution) to realistically looking images $G(z) \sim p_{fake}$. At the same time, a discriminator network D aims to classify well between samples coming from the real training data distribution $x \sim p_{real}$ and fake samples $G(z)$ generated by the generator. By utilizing the feedback of the discriminator, the parameters of the generator G can be adjusted such that its samples are more likely to fool the discriminator network in its classfication task. Mathematically speaking, the networks play a two-player minimax game against each other:

$$\min_{G} \max_{D} V(D,G) = \mathbb{E}_{x \sim p_{data}(x)}[log(D(x))] + \mathbb{E}_{z \sim p_z(z)}[1 - log(D(G(z)))] \qquad (1)$$

In consequence, as D and G are updated in an alternating fashion, the discriminator D becomes better in distinguishing between real and fake samples while the generator G learns to produce even more realistic samples, round by round.

**DCGAN**  The DCGAN architecture is a popular and well engineered convolutional GAN that is fairly stable to train and yields high quality results. The architecture is carefully designed with leaky relu activations to avoid sparse gradients and a specific weight initialization to allow for a robust training. It has proven to work reliably in the task of image synthesis at a resolution of 64x64px.

**LAPGAN**  The LAPGAN is a generative image synthesis framework inspired by the concept of Laplacian pyramids. Again, as seen in the standard GAN framework, a generator $G_0$ produces fake low resolution images $I_{0,fake}$ from noise $z_0 \sim p_z$. These images are then subject to an upsampling operation $up(\cdot)$ and fed, together with noise $z_1$, into the next generator $G_1$ of the pyramid, which is supposed to generate the fake residual image $R_{1,fake}$, i.e. the high frequency components which need to be added to the upsampled and thus blurry input image $I_{0,fake}$ to obtain a realistic, higher resolution image, i.e. $I_{1,fake} = up(I_{0,fake}) + R_{1,fake}$. The output is upsampled again and fed into the next higher resolution residual generator:

$$I_{0,fake} = G_0(z_0), \quad z_0 \sim p_z$$
$$I_{k,fake} = up(I_{k-1,fake}) + R_{k,fake}, \quad R_{k,fake} = G_k(up(I_{k-1,fake}), z_k), \quad k > 0$$

A peculiarity of this approach is the discrimination between real and fake residual images rather than the discrimination between real and fake images, i.e. a discriminator at level $k > 0$ operates on $R_{k,real}$ and $R_{k,fake}$ rather than $I_{k,real}$ and $I_{k,fake}$ (here referred to as *residual discrimination*). Interestingly, the framework by default is not trained end-to-end, even though theoretically possible. Instead, the different generators are trained separately, which makes the approach very time-consuming. Noteworthy, the LAPGAN has proven to work for synthesizing 96x96px sized images of realistically looking outdoor scenery, but to the best of our knowledge has not been applied to medical data yet.

## 3  Methodology

Like the LAPGAN, our DDGAN (Fig. 1) starts with a generator $G_0$ which maps noise $z_0 \sim p_z$ to low resolution image samples $I_0$. A respective low resolution discriminator then has to distinguish between real and fake images and provides the lowest resolution generator with gradients. In succession, the generated images are upsampled and fed into another generator. However, this generator differs from the LAPGAN generator in multiple aspects: First, opposed to the LAPGAN, the upsampled, generated images are not concatenated with another channel of noise. Second, any higher resolution generator $G_k, k > 0$ is simply a residual block(7) $res(\cdot, d)$ of depth $d$, whose output directly is an image $I_{k,fake}$, rather than a residual map (Fig. 2a). Consequently, and different to the LAPGAN, the discrimination happens on real and fake images rather than real and fake residuals (referred to as *image discrimination*). This way, the high frequency residuals that have to be added to

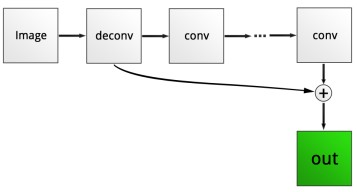

(a) Residual Deconvolution Block

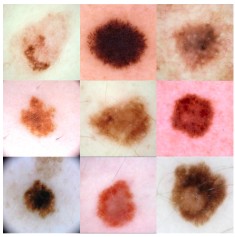

(b) Skin lesions in the ISIC 2017 dataset.

Figure 2

an upsampled version of the respective low resolution input image $I_{k-1,fake}$ are learned implicitly by the residual block.

$$I_{0,fake} = G_0(z_0), \quad z_0 \sim p_z$$
$$I_{k,fake} = G_k(up(I_{k-1,fake})) = res(up(I_{k-1,fake}), d), \quad k > 0$$

Further, the non-parametric upsampling can be replaced by a deconvolutional layer, which effectively amounts to learning an upsampling. Whether image discrimination is preferable over residual discrimination as seen in the LAPGAN, and if upsampling via deconvolution is somehow beneficial is subject to research in this manuscript.

## 4  Experiments and Results

In the first part of our experiments, we train a standard DCGAN, a LAPGAN and various DDGANs for skin lesion synthesis from the entire dataset and investigate the properties of the synthetic samples. In the second part, we utilize a selection of the frameworks to train synthesis models only on melanoma images in order to tackle class imbalance with the help of synthetic samples when training skin lesion classifiers.

### 4.1  Dataset

We evaluate our method on the ISIC 2017(3) dataset consisting of 2000 dermoscopic images of both benign and malignant skin lesions (images of 1372 benign lesions, 254 seborrheic keratosis samples and 374 melanoma). The megapixel dermoscopic images are center cropped and downsampled to $256 \times 256$px, leading to 2000 training images. Fig. 2b shows some of these training samples.

### 4.2  Evaluation Metrics

A variety of methods have been proposed for evaluating the performance of GANs in capturing data distributions and for judging the quality of synthesized images. In order to evaluate visual fidelity, numerous works utilized either crowdsourcing or expert ratings to distinguish between real and synthetic samples. There have also been efforts to develop quantitative measures to rate realism and diversity of synthetic images, the most prominent being the so-called Inception-Score, which relies on an ImageNet pretrained GoogleNet. Unfortunately, we noticed that it does not provide meaningful scores for skin lesions as the GoogleNet focuses on the properties of real objects and natural images. Odena et al.(15) rate sample diversity by computing the mean MS-SSIM metric among randomly chosen synthetic sample pairs, for which a high value indicates high sample diversity. Given the constrained nature of our images, this approach is also not applicable, since we obtain very high and comparable MS-SSIM values on the training dataset and on synthetic samples. Instead, per model we generate 2000 random samples, compute a normalized color histogram and compare it to the normalized color histogram of the training dataset in terms of the JS-Divergence and Wasserstein-Distance. Further, we discuss visual fidelity of the generated images with a focus on diversity, realism, sharpness and artifacts.

Table 1: Performance comparison of the DCGAN, LAPGAN and DDGAN. The models are compared in terms of the JS-Divergence and the Wasserstein Distance between the histogram of the training images and the histogram of samples generated using the respective model.

| Model | EMD | JS Divergence |
|---|---|---|
| DCGAN | 0.00821 | 0.00458 |
| LAPGAN | 0.04098 | 0.01420 |
| DDGAN$_{upsampling}$ | 0.02509 | 0.01099 |
| DDGAN$_{deconvolution}$ | 0.05410 | 0.02183 |

Figure 3: Histograms of the training dataset and of samples generated with different models

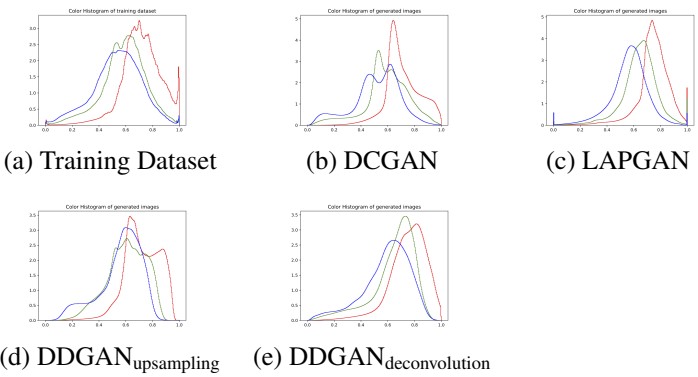

(a) Training Dataset      (b) DCGAN      (c) LAPGAN

(d) DDGAN$_{upsampling}$      (e) DDGAN$_{deconvolution}$

### 4.3 Image Synthesis

We trained a standard DCGAN, a LAPGAN and various DDGANs for skin lesion synthesis at a resolution of 256x256px. For a valid comparison, both the LAPGAN and the DDGAN are designed to have the same number of trainable parameters. Notably, the DCGAN directly regresses a single source of gaussian noise to images with a resolution of 256x256px, while LAPGAN and DDGAN increasingly regress from 64x64px to 128x128px up to 256x256px sized images. All of the models have been trained in an end-to-end fashion.

The dimensionality of $z_0$ is always set to $64$. As a loss function for the discriminator network we employ the least squares(12) loss. All models have been trained for 200 epochs, in minibatches of 8 due to GPU memory constraints on our nVidia 1080Ti, which took approx. 20h per model.

Overall, all of the models mimic the real data distribution fairly well (see Table. 1 and Fig. 4). Interestingly, the DCGAN matches the training dataset intensity distribution the best in all of the divergence measures, even though it shows the least sample variety and suffers from severe checkerboard artifacts. LAPGAN produces a great diversity of samples, but suffers from high frequency artifacts, as a result of high magnitude residuals. The DDGAN with standard upsampling and image discrimination matches the training dataset intensity distribution slightly better than the LAPGAN, but the sample diversity seems to be slightly less with any of the DDGAN models. Thereby, deconvolution seems to produce noisier and more unrealistic samples than standard upsampling.

### 4.4 Use-case Experiment

Next, we repeated the synthesis experiment, but trained models from only 374 melanoma images. Further, we split the ISIC dataset into a training (60% of the data) and validation set (40% of the data) while keeping the class distribution within each set, and utilized it to train a variety of skin lesion classifiers for classifying lesions into benign, melanoma and keratosis. Similar to the winner of the ISIC 2017 challenge(13), we utilize a pretrained RESNET-50 and train a baseline model $B_{Full}$ on the full training dataset, another baseline model $B_{Imb}$ where the number of melanoma images was artificially reduced to 46 to obtain a severe class imbalance, as well various RESNET-50, where

Figure 4: Samples generated with the different models.

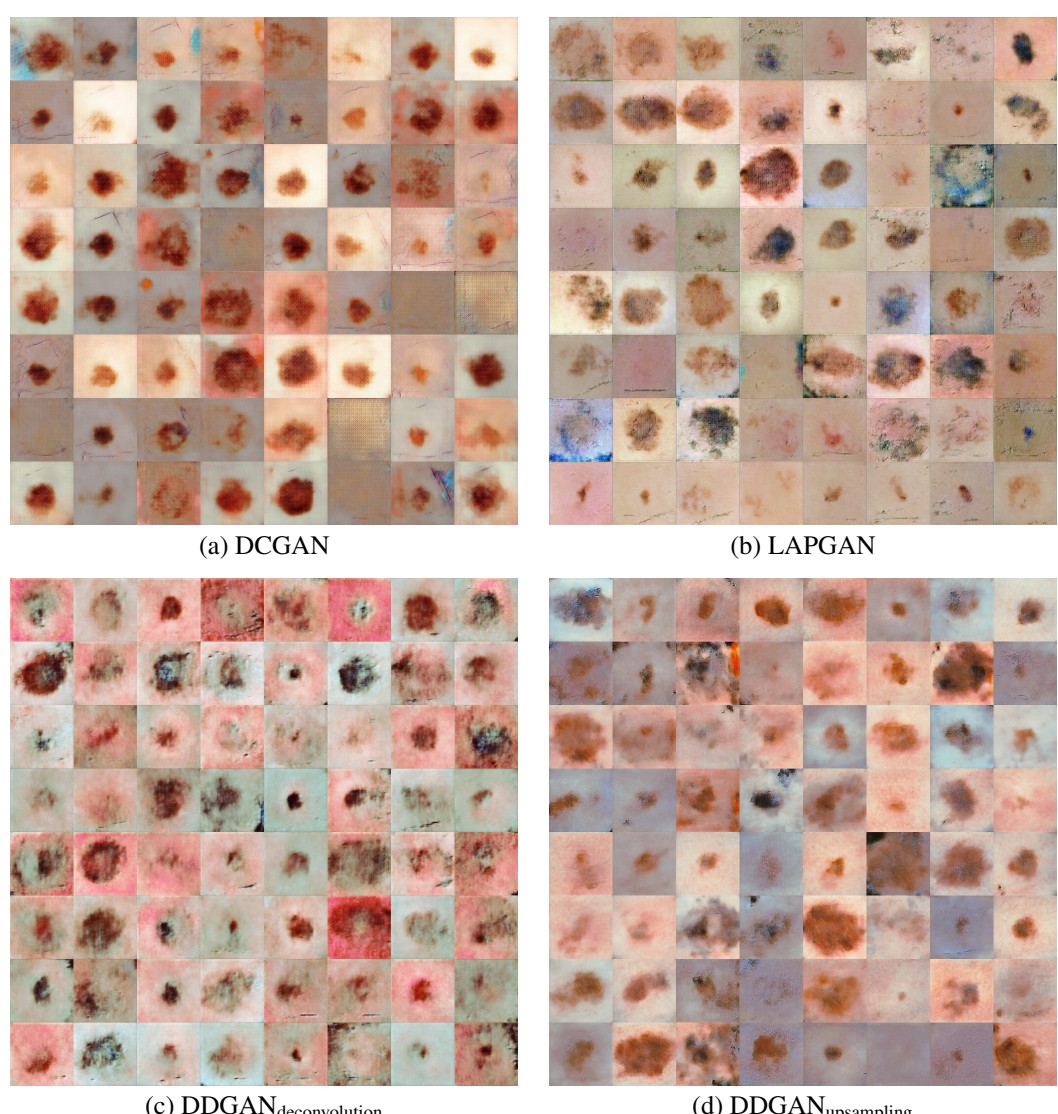

(a) DCGAN

(b) LAPGAN

(c) DDGAN$_{deconvolution}$

(d) DDGAN$_{upsampling}$

Table 2: Results of our use-case experiments, reporting the training and validation accuracy for different models.

| Set | B$_{Full}$ | B$_{Imb}$ | DCGAN | LAPGAN | DDGAN$_{upsampling}$ | DDGAN$_{deconvolution}$ |
|---|---|---|---|---|---|---|
| Training | 0.9809 | 0.8583 | - | **0.9929** | 0.9922 | 0.9914 |
| Validation | 0.7160 | 0.6394 | - | **0.7400** | 0.7268 | 0.7204 |

we recover the original class distribution in the training dataset by adding synthetic samples from the aforementioned synthesis models. All classifier models have been trained for 100 epochs on 224x224px sized images. Results are provided in Table. 2.

As we were unable to train a DCGAN from only 374 samples, it is not included in the Use-case experiment. For tackling class imbalance with synthetic samples we utilized the LAPGAN, the DDGAN$_{upsampling}$ and DDGAN$_{deconvolution}$.

As expected, in the presence of class imbalance, $B_{Imb}$ performs considerable worse than $B_{Full}$. Interestingly, when restoring the original class distribution of the training dataset with synthetic

samples we obtain even higher accuracies than with $B_{Full}$. Biggest improvements are made with samples from the LAPGAN, closely followed by DDGAN$_{upsampling}$ and DDGAN$_{deconvolution}$.

## 5   Discussion and Conclusion

In summary, we presented a comparison of the DCGAN, the LAPGAN and the DDGAN for the task of high resolution skin lesion synthesis and demonstrated that both the LAPGAN and the DDGAN are able to mimic the training dataset distribution with diverse and realistic samples, even when the training dataset is very small. In a set of use-case experiments, these synthetic samples have also been successfully used in the training of skin lesion classifiers for tackling class imbalance, even outperforming a baseline model purely trained on real data. We amount the observation that the histogram divergences are not consistent with synthesis quality of the models to the fact that the high frequency artifacts produced by the LAPGAN and the DDGAN bias the intensity histograms. This is reflected in the histogram obtained with DDGAN$_{upsampling}$ which comes closer to the training dataset histogram than the LAPGAN, as it produces less high frequency artifacts than the latter. Our qualitative and quantitative results have further shown that a learnt upsampling with the help of deconvolution is not superior to non-parametric upsampling. In our use-case experiments, the best performance was obtained with the LAPGAN, leaving us with the conclusion that having multiple sources of noise is indeed beneficial for realism and sample diversity. Interestingly, the high magnitude residual artifacts in LAPGAN do not seem to negatively impact the skin lesion classifier. Irrespective of that, we suppose that more training iterations might resolve these artifacts, but we also want to emphasize that training the LAPGAN is very difficult, requiring constant supervision and adjustment of hyperparameters. In comparison, the DDGAN with image discrimination is easier to train, converges faster and does not suffer from severe high frequency artifacts, while only being slightly inferior to the LAPGAN in our use-case experiments. In future work, we aim to obtain the feedback of dermatologists on sample realism, investigate the very recent approach for high resolution image synthesis presented in (9), and also conduct a variety of use-case experiments on data augmentation using synthetic samples.

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
