# OpenReview forum: "MelanoGANs: High Resolution Skin Lesion Synthesis with GANs"
_MIDL.amsterdam/2018/Conference — Submitted to MIDL 2018_

### Review · AnonReviewer3 · 2018-05-04
**The paper presents a relevant problem and a reasonable approach to solving it, but does not provide much analysis or experimental support for why the proposed method is better than the existing ones.**

**Rating:** 2
**Confidence:** 2

**Review:**

I think the paper has some merit, but there are weaknesses both in the argumentation and in the experimental results. I also think there may be a methodological flaw in the most important experiment (section 4.4).

Detailed review:

* The method is interesting and the use of image synthesis for data augmentation medical image analysis is relevant.

* The application to skin lesions is useful.

* In comparison with LAPGAN, the fact that DDGAN is easier to be trained end-to-end may be useful. (Although the experiments section states that the authors trained both models end-to-end.)

* The authors present the results of classification experiments, which is a reasonable practical use case for the synthetic samples. In these experiments, the synthetic examples improved the results when training with an imbalanced training set.

* The paper is generally easy to follow.

* Analytically, the paper is not very strong. It presents a new DDGAN method and explains the differences with LAPGAN and DCGAN. However, it does not really try explain why these differences would be beneficial. I think the paper could have made a stronger case for these design decisions.

* Similarly, the discussion of the results is mostly limited to a dry presentation of the performance estimates. The paper tells us DDGAN works (although not clearly better than LAPGAN, DCGAN), but it does not really explain how this is a logical result of the design.

* The first part of the results evaluates the synthetic samples directly. I understand that this is difficult, but I don't find the results convincing. From Figure 3 and 4 I would not directly conclude that DDGAN works better than the other methods. Comparing histograms seems to be a very crude way to compare the results.

* In the Discussion, the argument that DDGAN produces fewer high-frequency artifacts than LAPGAN is a bit strange. This might make the synthetic samples nicer to look at, but does that make them *better* synthetic samples? The authors tried this in their classification experiments, but found that the LAPGAN samples were more useful than than the DDGAN samples. The suggestion that the LAPGAN samples might have been better if it had had more training iterations is an honest observation, but doesn't make the argument stronger.

* I like the suggestion for future work in which the authors would like to ask dermatologists to provide feedback on realism. This would certainly be interesting, but I would still doubt if that says anything about how useful the samples are. Unless the images are only generated to be viewed by people, realism doesn't say everything.

* In this regard, the use case experiment with data augmentation in section 4.4 seems more informative, because it allows us to objectively measure an actual outcome, on a task for which the synthesis model might help. Unfortunately, it appears that LAPGAN works a little bit better than the two DDGAN methods, so this also does not provide a strong support for DDGAN.

* Moreover, I have some doubts about the experimental design of the experiments in section 4.4. I get the impression that to train the synthesis models in the experiment in section 4.4, the authors used the full dataset and not just the 60% training images (because they refer to 364 melanoma samples, which corresponds to the number given for the full dataset). If they did indeed do this, I think this would be a problem: by training the synthesis model on images from the full set, it would be very well possible that the synthesis memorises and can reproduce some of the test images. For a proper investigation of how the GANs do in a data augmentation task, I think the synthesis models would have to be trained on only the available set of 46 melanoma images. (It is possible that the authors used only the 46 images, but the paper suggests that they did not.)

* The argument that DDGAN is easier to train than LAPGAN sounds like a good reason to use it. However, I think the paper never really explains why DDGAN is easier to train than LAPGAN, nor does it provide numerical arguments that support this argument (e.g., if the authors state that DDGAN converges faster, it would be helpful if they included a plot that illustrates this).


Minor comments:

* For a final version, I would suggest that the authors clean up the references. It looks a bit sloppy: incorrect capitalisation, missing conference names. There is also a fairly large number of references to arXiv papers that have since been published. I think it would be better to cite the official version if it is available.

* Table 1: I would use the same ordering of JS-Divergence and Wasserstein distance in the caption and in the table. It is also confusing that the Wasserstein distance is listed as "EMD" in the table.

**Special Issue:**

No

---

### Review · AnonReviewer1 · 2018-05-07
**Author presented a new GAN architecture for synthesize high resolution realistic melanoma images. Unlike related synthesis work using GAN, this work is one of the first attempts to synthesize higher resolution medical images. Similar to LAPGAN the proposed DDGAN network followed the pyramidal network structure, but the discriminator is performed on images rather than residuals. Besides, the proposed DDGAN enable an end-to-end training which shortens the training time**

**Rating:** 1
**Confidence:** 2

**Review:**

The writing could be improved. There exists many format errors, grammar mistakes and confusing graph and tables. The authors did not give any conclusive evidence that the proposed image-based discriminator has any advantage compared to residuals-based discriminator. The experimental results also did not show much promising results comparing to related approaches.
One of the main contributions is synthesis of high resolution realistically-looking images. However, the max resolution in this work is 256*256px. I think this resolution can also be handled by other previously proposed GAN architectures. The original resolution of ISIC data is 500*500px. Why do not synthesize to original resolution?

Detailed comments:

1. Author enumerated several inapplicable evaluation metrics for this work, and used EMD, JSD and histogram in this paper. However, for a more comprehensive work, I recommend to perform an experiment on phantom data or natural images using common evaluation metrics, if this network is also applicable for other modalities rather than melanoma images. In addition, the authors did not give any quantitative measure for the histograms, visual measure did not give much meaningful information for evaluation.

2. It would have been better to give detailed comparisons of the real and fake melanoma images. The visual fidelity of DDGAN's results looks smoother and more realistic than other methods but it does not indicate DDGAN's results is more similar to real melanoma GAN. Besides, the visual fidelity seems to be in conflict with quantitative evaluations.

3. Although the authors gave use-case experiments to show that synthesis images can help to tackle to class imbalance problem, the experiment does not seem to have a close relation with the proposed network. Instead, the underperformance confirmed its weakness again.


**Special Issue:**

No

---

### Review · AnonReviewer2 · 2018-05-13
**An interesting idea but badly written**

**Rating:** 2
**Confidence:** 3

**Review:**

Overall:
The paper copes with a problem of imbalanced data phenomenon in medical imaging. The authors propose to use data augmentation by applying Generative Adversarial Networks (GANs). The presented idea is verified on a skin lesion classification. However, the paper is written in a chaotic manner, many concepts are not properly described. Moreover, the experiment lacks many details which makes it difficult to follow. The paper has a potential, however, it is not ready in its current form.

Strengths:
+ The paper tries to handle imbalanced data phenomenon that is an important issue in medical imaging.
+ The proposed approach seems to be interesting and has a potential.

Remarks:
* Major
- The idea of using multiple discriminators is not new and was already proposed, e.g.:
Durugkar, I., Gemp, I., & Mahadevan, S. (2016). Generative multi-adversarial networks. arXiv preprint arXiv:1611.01673.
- The experiment section is written in a chaotic manner. First, Section 4.2 is not convincing. The authors claim tha all models fit the data distribution. However, how to read the results in Table 1? What is a score for a "bad" model? Some baseline is definitely necessary. Second, the main point of the paper is the data augmentation and the subsection 4.4 is critical for the paper. However, the experiment is not precisely described. Were GANs trained on full dataset or on the imabalanced data? If they were trained on the full data, then the results are meaningless.
- The paper is hard to read. Many descriptions of important concepts are missing which makes the paper rather chaotic.

* Minor
- LAPGAN and DCGAN are useful models but these are not current state-of-the-architectures. However, the authors make such statement in the contribution paragraph.

**Special Issue:**

No

---

### Decision · Program_Chairs · 2018-05-15
**Paper76 Acceptance Decision**

Reject